# Somatic Mutation Profile as a Predictor of Treatment Response and Survival in Unresectable Pancreatic Ductal Adenocarcinoma Treated with FOLFIRINOX and Gemcitabine Nab-Paclitaxel

**DOI:** 10.3390/cancers16152734

**Published:** 2024-08-01

**Authors:** Rodrigo Paredes de la Fuente, Santiago Sucre, Cristina Ponce, Ahmed Anwer Ali Rattani, Mary Linton B. Peters

**Affiliations:** 1Department of Medicine, Icahn School of Medicine at Mount Sinai, New York, NY 10029, USA; 2Division of Medical Oncology, Department of Medicine, Beth Israel Deaconess Medical Center, Boston, MA 02215, USAarattani@bidmc.harvard.edu (A.A.A.R.)

**Keywords:** pancreatic ductal adenocarcinoma, molecular profiling, precision oncology, treatment response, chemotherapy

## Abstract

**Simple Summary:**

This study explored how genetic changes influence treatment effectiveness and survival in patients with advanced pancreatic cancer. By examining tumors from 142 patients, researchers identified specific genetic mutations that can predict how well a patient responds to chemotherapy. Findings showed that mutations in certain genetic pathways are associated with longer survival and better treatment outcomes. This research highlights the importance of tailoring cancer treatment based on individual genetic profiles, potentially leading to more personalized and effective therapies for pancreatic cancer patients.

**Abstract:**

(1) Background: Pancreatic ductal adenocarcinoma (PDAC) has low survival rates despite treatment advancements. Aim: This study aims to show how molecular profiling could possibly guide personalized treatment strategies, which may help improve survival outcomes in patients with PDAC. (2) Materials and Methods: A retrospective analysis of 142 PDAC patients from a single academic center was conducted. Patients underwent chemotherapy and next-generation sequencing for molecular profiling. Key oncogenic pathways were identified using the Reactome pathway database. Survival analysis was performed using Kaplan–Meier curves and Cox Proportional Hazards Regression. (3) Results: Patients mainly received FOLFIRINOX (n = 62) or gemcitabine nab-paclitaxel (n = 62) as initial chemotherapy. The median OS was 13.6 months. Longer median OS was noted in patients with NOTCH (15 vs. 12.3 months, *p* = 0.007) and KIT pathway mutations (21.3 vs. 12.12 months, *p* = 0.04). Combinatorial pathway analysis indicated potential synergistic effects on survival. In the PFS, PI3K pathway (6.6 vs. 5.7 months, *p* = 0.03) and KIT pathway (10.3 vs. 6.2 months, *p* = 0.03) mutations correlated with improved PFS within the gemcitabine nab-paclitaxel subgroup. (4) Conclusions: Molecular profiling could play a role in PDAC for predicting outcomes and responses to therapies like FOLFIRINOX and gemcitabine nab-paclitaxel. Integrating genomic data into clinical decision-making can benefit PDAC treatment, though further validation is needed to fully utilize precision oncology in PDAC management.

## 1. Introduction

Pancreatic cancer presents a significant challenge as it is the third-leading cause of cancer-related deaths [1], with a rising incidence that is expected to make it the second-leading cause by 2030 [2]. Despite advancements in therapy, the 5-year survival rate remains low at 11.5% [3]. While surgical resection offers a curative option, it is feasible for only a small percentage of patients because this cancer often presents at an advanced stage, limiting the potential for curative interventions. [4,5]. Despite extensive study, immune–oncology agents have had little benefit in pancreatic ductal adenocarcinoma (PDAC), and targeted treatment based on molecular pathological tumor characteristics has yet to be as broadly beneficial as they have been in other tumor types [6]. Aggressive cytotoxic chemotherapy remains the primary treatment option for unresectable pancreatic tumors, despite its relatively low response rate [7]. Treatment regimens for PDAC vary based on the patient’s performance status [8]. FOLFIRINOX, which includes infusional 5-fluorouracil, leucovorin, irinotecan, and oxaliplatin, is preferred for patients with good performance status (ECOG 0-1) due to its superior efficacy compared to gemcitabine alone. For patients with a slightly lower performance status (ECOG 1-2), the combination of gemcitabine plus nab-paclitaxel is another frontline option.

PDAC exhibits significant molecular heterogeneity, with an average of from 60 to 70 genetic changes observed [9]. Efforts have been made to categorize pancreatic cancer based on molecular activity and phenotypical aspects. However, due to the complexity of the disease, a consensus has not been reached [10,11,12,13,14]. Nonetheless, profiling studies have identified potentially actionable molecular alterations in up to 25% of pancreatic cancers [15]. The advent of next-generation sequencing (NGS) has revolutionized genomic profiling and made it widely accessible, and it plays a crucial role in pancreatic cancer research and clinical decision-making [16,17,18]. Leveraging NGS technology can lead to a more comprehensive understanding of the molecular landscape of pancreatic cancer, paving the way for improved treatments and future advancements in the field [11].

In this study, we retrospectively examined the molecular profiles of unresectable PDAC patients, aiming to elucidate the relationship between specific genetic alterations, natural history of disease, and treatment responses. Our focus was on understanding how the presence of mutations in key oncogenic pathways might influence the overall survival and efficacy of first-line chemotherapy regimens, such as FOLFIRINOX and gemcitabine nab-paclitaxel. By integrating molecular data with clinical outcomes, this study seeks to contribute to the burgeoning field of precision oncology, offering insights into personalized treatment strategies that could potentially improve survival and quality of life for patients with this formidable disease.

## 2. Materials and Methods

This study is a retrospective chart review designed to investigate the role of molecular profiling in predicting treatment response and survival in patients with unresectable pancreatic ductal adenocarcinoma.

Study Population: The study population consisted of patients diagnosed with PDAC who presented to the medical oncology clinic of a single center (Beth Israel Deaconess Medical Center, Boston MA) between the years 2013 and 2022. Only those patients with next-generation tumor sequencing who underwent cytotoxic chemotherapy were selected for inclusion. The sample size was calculated to ensure a power of 0.80 and a significance level of 0.05. Patients who were included followed these characteristics: diagnosed with unresectable PDAC (Stage III or IV) [19], underwent NGS for molecular profiling, and received first-line systemic cytotoxic chemotherapy (FOLFIRINOX or gemcitabine nab-paclitaxel). Of those, we obtained a sample of 142 patients.

Data Collection: A targeted chart review was performed which captured demographic information, age at diagnosis, performance status at diagnosis, tumor histology, and first-line chemotherapy regimen. Although we captured all first-line chemotherapy regimens, most patients received standard-of-care treatment with either FOLFIRINOX [20,21] or gemcitabine nab-paclitaxel [20,22]. Disease progression was assessed using the Response Evaluation Criteria in Solid Tumors (RECIST) criteria, as determined by provider documentation [23].

Molecular Profiling: The genetic profile of each tumor sample was obtained through next-generation tumor sequencing using the Foundation Medicine CDX Report [24]. This analysis includes 324 genes that are potentially cancer-related, of which 225 were altered in at least one sample within our selected patient population. These genes were grouped into 14 different pathways (Appendix A) based on their biological function using the Reactome Pathway database [25]; some genes may be members of multiple pathways. Each patient sample was classified by the altered genes present, and then by the pathways represented by those mutations. We recorded whether mutations were present in these pathways as a binary yes-or-no variable. Any insertion, deletion, substitution, duplication, rearrangement, truncation, or amplification was considered a mutated state [26]. All data were collected and managed using REDCap electronic data capture hosted at Beth Israel Deaconess Medical Center [27,28]. This study was approved by the BIDMC IRB and determined to be exempt from informed consent requirements.

Statistical Analysis: We conducted two distinct survival analyses based on the presence or absence of specific mutated pathways: overall survival and progression-free survival. We performed a sub-group analysis of the patients who received FOLFIRINOX or gemcitabine nab-paclitaxel as a first-line therapy. We used Kaplan–Meier curves and log-rank tests to compare survival between groups with and without each specific mutated pathway. We also performed a Cox Proportional Hazards Regression analysis to adjust for potential confounding variables. The threshold for statistical significance was set at *p* < 0.05. All analyses were performed using R Statistical Software (v4.2.2; R Core Team 2021) via the survival package [29,30].

## 3. Results

### 3.1. Characteristics of the Patient Population

This study included 142 patients with a median age of 66 years who were balanced in gender (49% male, 51% female). Most patients were diagnosed at advanced stages (37% Stage III, 63% Stage IV). The majority underwent standard first-line chemotherapy, with equal distribution between FOLFIRINOX (44%) and gemcitabine nab-paclitaxel (44%) (Table 1).

### 3.2. Characteristics of the Pathology Samples

The majority of samples were tissue samples obtained from the primary tumor (68, 48%) or a metastatic site (68, 48%), while a small number were from peripheral blood (6, 4%). Tumor mutation burden (TMB) status was known for 110 patients, with a median value of 2.5 (IQR 1.3–3.8), and 4 patients classified as TMB-High. MSI status was known for 140 patients, with 1 classified as MSI-H.

The comprehensive genomic analysis of the 142 PDAC patient samples revealed a significant degree of mutational burden, with an average of 7.83 mutations per sample, ranging from 1 to 35 mutations. Predominantly, mutations were observed in the MAPK-RAS pathway in 94% of patients (n = 134) and the TP53 pathway in 87% (n = 123). Other frequently altered pathways included the Cell Cycle (CC) pathway in 76% of patients (n = 108), DNA Damage Repair (DDR) in 49% (n = 70), TGF-beta pathway in 46% (n = 65), PI3K pathway in 37% (n = 52), and NOTCH pathway in 36% (n = 51). Less commonly altered pathways comprised FGFR (23%, n = 32), WNT (20%, n = 29), PDGFR (18%, n = 25), KIT (16%, n = 23), ERBB2 (14%, n = 20), ALK (13%, n = 19), and FTL3 (6%, n = 9). These findings highlight the extensive molecular diversity present in PDAC and underscore the potential for targeted therapeutic strategies based on individual mutational profiles.

### 3.3. Overall Survival Analysis

In our study cohort, the median overall survival (OS) was found to be 13.6 months. A detailed analysis of the genetic profiles revealed significant survival advantages in specific mutational pathways. Notably, patients exhibiting mutations in the NOTCH pathway (n = 51) demonstrated a median OS of 15 months, markedly longer than the 12.3 months observed in those without NOTCH mutations (*p* = 0.007; HR 0.57; 95% CI 0.38–0.86) (Figure 1A). This suggests a potential protective effect of NOTCH pathway alterations on survival.

Furthermore, a similar trend was observed in the KIT pathway mutations. Among the 23 patients with KIT pathway mutations, the median OS was significantly extended to 21.3 months, compared to 12.12 months in patients without these mutations (*p* = 0.04; HR 0.59; 95% CI 0.35–0.98) (Figure 1B). This finding suggests a notable survival benefit associated with KIT pathway alterations.

In contrast, other individual pathways did not show a statistically significant association with OS in our cohort. However, an interesting non-significant trend towards improved OS was noted in patients with mutations in the ALK pathway, where the median OS was 16.7 months compared to the overall cohort median of 13.6 months (*p* = 0.06, HR = 0.57, 95% CI 0.32–1.03). This observation warrants further investigation to elucidate the potential role of ALK pathway mutations in PDAC survival.

We further tested all two-way combinations of mutations among the 14 pathways. In the two-way combination analysis (Table 2 and Figure 2), mutations in the ALK, KIT, ERBB2, and PDGFR pathways could be synergistic with NOTCH, showing increases in overall survival (median OS of 25, 24, 25, and 23 months vs. 15 months for NOTCH overall). Combinatoric analysis with the KIT pathway revealed a potentially negative effect of combination with the DDR pathway (median OS of 11 months vs. 21 months for KIT overall). Although the ALK pathway overall did not show a statistically significant increase in OS, in combination with TGFB (median OS 28 months) and TP53 (median OS 24 months) it did achieve significance in the combination analysis.

### 3.4. Progression-Free Survival Analysis

In our study, we examined the progression-free survival (PFS) among the 142 patients with PDAC, specifically analyzing differences in PFS based on first-line chemotherapy regimens and genetic mutations. The median PFS for patients treated with FOLFIRINOX (n = 62) was 9.1 months, while those receiving gemcitabine nab-paclitaxel (n = 62) had a median PFS of 6.3 months.

For the FOLFIRINOX cohort, the analysis did not reveal any significant correlation between the presence of specific mutational pathways and improvements in PFS. Even with the exploration of potential two-way interactions between different pathways, no statistically significant results emerged that suggested a benefit in PFS from combined mutations in the FOLFIRINOX-treated group.

In contrast, the patients treated with gemcitabine nab-paclitaxel exhibited notable associations between certain genetic alterations and PFS. Notably, the presence of a PI3K pathway mutation (found in 25 patients) was associated with a significantly extended PFS of 6.6 months, compared to 5.7 months in patients without such mutations (*p* = 0.03; HR = 0.54, 95% CI 0.31–0.93). Furthermore, mutations in the KIT pathway (observed in 10 patients) correlated with an even more substantial increase in PFS, reaching 10.3 months versus 6.2 months in non-mutated cases (*p* = 0.03; HR 0.42, 95% CI 0.2–0.91). Additionally, mutations in the FTL3 pathway (n = 4) were linked to the longest median PFS of 13.3 months, a significant improvement compared to 6.2 months in those without FTL3 mutations (*p* = 0.02; HR 0.21, 95% CI 0.05–0.89).

These results highlight the potential of certain genetic mutations, particularly within the PI3K, KIT, and FTL3 pathways, to predict longer progression-free survival in PDAC patients receiving gemcitabine nab-paclitaxel as their first-line treatment. (Figure 3) This insight underscores the value of molecular profiling in guiding therapy choices and tailoring treatment strategies to individual patient profiles.

In a two-way combination analysis, these three pathways (PI3K, KIT, and FTL3) could be synergistic for response to gemcitabine nab-paclitaxel, with a longer PFS in combination (PI3K + KIT 10.9 months, PI3K + FTL3 13.1 months, KIT + FTL3 19.7 months). In addition, DDR pathway mutations in combination with KIT or FTL3 (but not PI3K) appeared to provide additional PFS benefits (DDR + KIT 14.7 months, DDR + FTL3 19.7 months) (Table 3 and Figure 4).

## 4. Discussion

Our study aimed to comprehensively investigate the molecular profiles and clinical characteristics of patients with PDAC. We placed particular emphasis on identifying potential predictive biomarkers and evaluating their significance in treatment selection and outcomes. This analysis consisted of 142 PDAC patients, with a relatively equal gender distribution and a median age of 66 years. In line with current treatment guidelines, the most frequently administered first-line chemotherapy regimens in our cohort were gemcitabine nab-paclitaxel and FOLFIRINOX [20]. Within our study, a small subset of patients exhibited TMB-H or MSI-H characteristics, a known characteristic that contributes to the limited efficacy of immunotherapy in pancreatic cancer [31,32,33,34].

In the current study, tumors with NOTCH-mutated pathways exhibited a longer OS compared to tumors without mutations in the NOTCH pathway. The role of NOTCH signaling in pancreatic cancer is complex and context-dependent. It appears to have both oncogenic and tumor-suppressive functions, with different NOTCH receptors playing distinct roles [35,36,37]. Individual NOTCH receptors can exhibit opposing activities and lead to diverse cellular responses [38]. Furthermore, NOTCH signaling is intricately intertwined with other signaling pathways and mechanisms, further complicating our understanding of its involvement in oncogenic processes [39,40]. The NOTCH signaling pathway has emerged as a crucial regulator in triggering epithelial-to-mesenchymal transition (EMT) [41], a transformative process that intensifies tumor aggressiveness by enhancing cellular motility and metastatic potential [42]. In addition, the NOTCH pathway has been linked to chemotherapy resistance in tumors [43,44], and therapeutic interventions targeting the NOTCH signaling pathway have shown potential for reversing chemoresistance [45]. We did not see a PFS difference between groups in our study. Interestingly, in our cohort, this improvement in OS with NOTCH-mutated pathways persisted even when combined with other mutated pathways such as TP53, which are typically associated with poor prognosis [46,47].

The consistent superior performance of NOTCH-mutated pathways raises intriguing questions. Some studies have proposed that highly mutated NOTCH samples exhibit significantly enhanced immunogenicity and immune response [48]. Additionally, other studies have suggested that the specific genes mutated within the NOTCH pathway may also play a role in determining prognosis [49]. The therapeutic targeting of the NOTCH pathway has been investigated in various trials using medications that globally inhibit the pathway [50,51]. However, other studies [52,53] suggest that the therapeutic targeting of the NOTCH pathway may be more complex than initially believed. The prognostic and predictive implications of NOTCH pathway mutations, as well as the potential targeting of this pathway for therapeutic benefit, will be a topic of substantial interest.

In our cohort, we identified a more favorable OS and PFS in samples exhibiting mutations within the KIT pathway. Furthermore, our findings indicate a synergistic effect as the combined presence of NOTCH and KIT mutations was associated with significantly prolonged OS compared to the analysis of these mutations individually. While the existing knowledge on pancreatic cancer is limited, certain studies have suggested that KIT may serve as a valuable prognostic marker for favorable outcomes in specific solid tumors [54]. However, the majority of available evidence suggests KIT as a marker for poorer prognosis [55,56]. Conversely, ALK has been proposed as a potential biomarker for favorable prognosis in lung cancer [57]. Although our results align with this finding, the understanding of ALK’s role in pancreatic cancer prognosis remains constrained due to limited data. Lastly, we observed that PI3K mutations were associated with improved PFS compared to samples lacking such mutations. Despite disappointing evidence regarding its therapeutic targeting [58], PI3K has been shown to interact with other pathways and influence survival outcomes [59].

Finally, this analysis highlights the potential role of chemotherapy selection, specifically noting that gemcitabine nab-paclitaxel may exhibit better performance in the presence of certain mutation pathways. Platinum-based therapy has been recognized for its effectiveness in tumors with homologous combination defects [60]. However, there is limited clinical data available for gemcitabine or taxanes in this context [61]. To predict in vivo tumor response to treatment, several studies have utilized PDX and organoid models [62], where the mutational profile serves as a readily available tool. This hypothesis-generating information suggests that mutational profile analysis should be considered for inclusion in future clinical trials.

This study has several limitations that should be taken into account when interpreting the findings. Firstly, the retrospective study design introduces inherent limitations. Data collected retrospectively from chart reviews may be influenced by biases and limitations in data availability, quality, and completeness. Secondly, this study’s single-center nature restricts the generalizability of the results. Moreover, this study’s sample size is relatively small, and the presence of missing data for OS or PFS analysis, as well as the subdivision of patients by treatment, further reduced the sample size. This limited sample size may affect the statistical power and precision of the findings. Finally, it is important to acknowledge that the genetic analysis was based on pathways that grouped several genetic mutations, thus limiting the scope of the analysis. This approach may have overlooked important genetic alterations that could specifically impact treatment response and survival. A more comprehensive and broader genetic profiling approach may provide additional insights into the study’s subject matter.

## 5. Conclusions

These findings contribute to our understanding of the clinical characteristics and treatment patterns of pancreatic cancer patients and highlight the potential for personalized treatment approaches based on molecular profiling. The integration of molecular profiling, such as next-generation sequencing, holds great promise for identifying actionable targets and guiding personalized therapeutic approaches in PDAC. However, further studies are needed to validate these findings and determine their clinical implications.

## Figures and Tables

**Figure 1 cancers-16-02734-f001:**
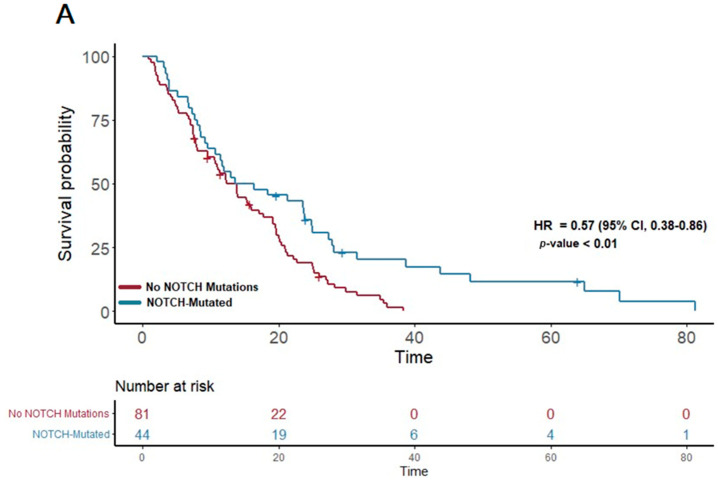
(**A**) The first panel shows overall survival (OS) for patients with NOTCH mutations (median OS: 15 months) versus those without (median OS: 12.3 months) (*p* = 0.007; HR 0.57; 95% CI 0.38–0.86). (**B**) The second panel shows OS for patients with KIT mutations (median OS: 21.3 months) versus those without (median OS: 12.12 months) (*p* = 0.04; HR 0.59; 95% CI 0.35–0.98).

**Figure 2 cancers-16-02734-f002:**
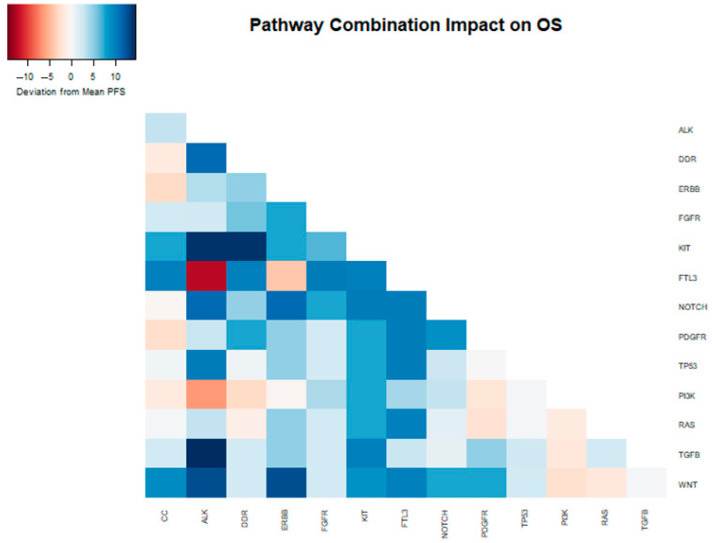
Heatmap depicting the deviation from the median OS (13.6 months) within the entire sample, attributed to the presence of distinct mutations and their combinations in the analyzed samples.

**Figure 3 cancers-16-02734-f003:**
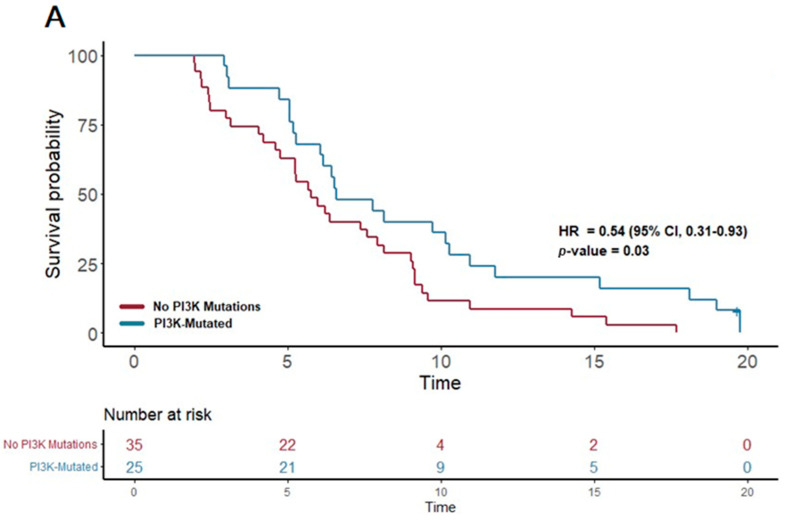
Kaplan–Meier plots for PFS in patients that received first-line therapy gemcitabine nab-paclitaxel. (**A**) The first panel shows PFS for patients with PI3K mutations (median PFS: 6.6 months) versus those without (median PFS: 5.7 months) (*p* = 0.03; HR 0.54; 95% CI 0.31–0.93). (**B**) The second panel shows PFS for patients with KIT mutations (median PFS: 10.3 months) versus those without (median PFS: 6.2 months) (*p* = 0.03; HR 0.42; 95% CI 0.2–0.91).

**Figure 4 cancers-16-02734-f004:**
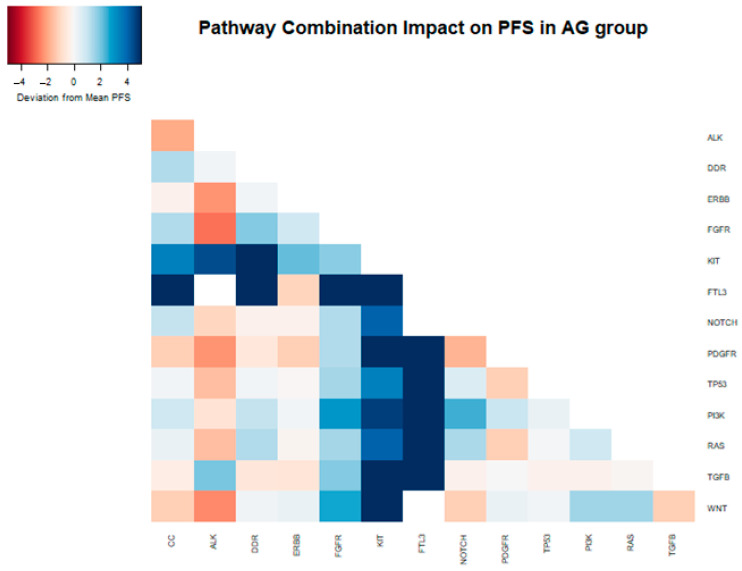
Heatmap depicting the deviation from the median PFS (6.29 months) within the samples of patients that received gemcitabine nab-paclitaxel as their first-line therapy, attributed to the presence of distinct mutations and their combinations in the analyzed samples.

**Table 1 cancers-16-02734-t001:** Demographic and sample characteristics.

Patient Characteristics	
Age, years (median, IQR)	66 (59–72)
Gender	
Male	70 (49%)
Female	72 (51%)
ECOG	
0	83 (58%)
1	43 (30%)
2	10 (8%)
3	3 (2%)
Unknown	3 (2%)
Stage	
III	52 (37%)
IV	90 (63%)
First-line Chemotherapy	
FOLFIRINOX	62 (44%)
Gemcitabine nab-paclitaxel	62 (44%)
FOLFOX	8 (6%)
Gemcitabine monotherapy	6 (4%)
Gemcitabine plus Cisplatin	2 (1%)
Other cytotoxic chemotherapy	2 (1%)
Sample Characteristics	
Source	
Primary	68 (48%)
Metastasis	68 (48%)
Blood	6 (4%)
TMB (median, IQR)	2.5 (1.3–3.8)

**Table 2 cancers-16-02734-t002:** Overall survival of mutations in patient samples and combination analysis.

Main Pathway	Mutation Combination	n	Median OS (months)	HR	95% CI	*p*-Value (HR)
Overall		142	13.6			
NOTCH		51	15.0	0.57	0.38–0.86	0.008
	NOTCH + CC	38	13.3	0.59	0.38–0.91	0.017
	NOTCH + PI3K	18	16.7	0.57	0.32–1	0.052
	NOTCH + ALK	11	24.9	0.39	0.19–0.81	0.011
	NOTCH + KIT	14	23.8	0.43	0.23–0.81	0.009
	NOTCH + DDR	29	18.4	0.60	0.38–0.96	0.030
	NOTCH + ERBB2	11	24.9	0.42	0.2–0.86	0.018
	NOTCH + PDGFR	14	22.5	0.41	0.21–0.8	0.009
	NOTCH + TP53	39	16.4	0.53	0.34–0.82	0.004
	NOTCH + RAS	42	15.0	0.66	0.45–0.99	0.044
KIT		23	21.3	0.59	0.35–0.98	0.043
	KIT + CC	19	21.3	0.56	0.33–0.96	0.035
	KIT+ DDR	13	27.8	0.49	0.26–0.92	0.028
	KIT + TP53	20	21.3	0.57	0.33–0.96	0.033
	KIT + PI3K	14	21.3	0.46	0.24–0.87	0.017
	KIT + RAS	21	21.3	0.59	0.35–0.98	0.043
ALK		19	16.7	0.57	0.32–1.03	0.064
	ALK + CC	14	16.7	0.53	0.28–1	0.050
	ALK + TGFB	7	28.2	0.34	0.14–0.85	0.021
	ALK + TP53	15	23.8	0.52	0.28–0.96	0.037

**Table 3 cancers-16-02734-t003:** Progression-free survival of mutations and combination analysis in patient samples with gemcitabine nab-paclitaxel as first-line therapy.

Main Pathway	Mutation Combination	n	Median PFS (months)	HR	95% CI	*p*-Value (HR)
Overall		62	6.3			
PI3K		25	6.6	0.54	0.31–0.93	0.027
	PI3K + CC	20	7.2	0.53	0.29–0.95	0.034
	PI3K + DDR	16	7.4	0.47	0.24–0.9	0.023
	PI3K + FGFR	12	9.2	0.45	0.22–0.9	0.024
	PI3K + KIT	7	10.9	0.32	0.13–0.83	0.018
	PI3K + FTL3	4	13.1	0.21	0.05–0.89	0.035
	PI3K + TP53	23	6.6	0.55	0.31–0.97	0.037
	PI3K + RAS	24	7.2	0.52	0.3–0.91	0.022
KIT		10	10.3	0.42	0.2–0.91	0.028
	KIT + CC	9	9.7	0.43	0.19–0.98	0.046
	KIT + DDR	6	14.7	0.26	0.09–0.74	0.012
	KIT + FTL3	3	19.7	0.11	0.01–0.83	0.032
	KIT + TP53	9	9.7	0.43	0.19–0.97	0.043
	KIT + RAS	10	10.3	0.42	0.2–0.91	0.028
FTL3		4	13.1	0.21	0.05–0.89	0.035
	FTL3 + CC	4	13.3	0.21	0.05–0.89	0.035
	FTL3 + DDR	3	19.7	0.11	0.01–0.83	0.032
	FTL3 + FGFR	3	19.7	0.11	0.01–0.83	0.032
	FTL3 + TP53	4	13.1	0.21	0.05–0.89	0.035
	FTL3 + RAS	4	13.3	0.21	0.05–0.89	0.035

## Data Availability

The data supporting the findings of this study are available from the corresponding author, Mary Linton B. Peters, upon reasonable request. The data are not publicly available due to privacy or ethical restrictions.

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
