# Peer review of "Somatic Mutation Profile as a Predictor of Treatment Response and Survival in Unresectable Pancreatic Ductal Adenocarcinoma Treated with FOLFIRINOX and Gemcitabine Nab-Paclitaxel"

_cancers, 2024, doi:10.3390/cancers16152734_

Round 1

Reviewer 1 Report

Comments and Suggestions for Authors

Title requires correction(s)

Methodological Biases exist (eg. How did the Authors determine the study size,

Inclusion/Exclusion Criteria, was the questionnaire standardized???,etc.)

Some Refs are missing

(The Authors must see my remarks)

Author Response

Response to Reviewer X Comments

Thank you for your thorough review and valuable suggestions. We appreciate your feedback, which has been instrumental in enhancing the quality of our manuscript. Please find our detailed responses below, along with the corresponding revisions and corrections highlighted in the re-submitted files.

  1. Point-by-point response to Comments and Suggestions for Authors

Commented [M1]: Please state the type of the arti-cle, eg. Research????

Author Response 1: Thank you for pointing this out. We added "Research Article"

Commented [M2]: State the drugs (chemotherapy) examined....

Author Response 2: We agreed with your comment and corrected the title corrected for "Commented [M2]: State the drugs (chemotherapy) examined...."

Commented [M3]: Add "/Aim"....

Response 3: We agree with this comment. We added Aim in line 23.

Commented [M4]: Replace by "Materials and Meth-ods"....

Response 4: We replaced "Materials and Methods" in line 75

Commented [M5]: Here, you must write the full-name of the disease....

Response 5: Thanks you for pointing this out. We have corrected this issue in line 50.

Commented [M6]: Use the abbreviation of the dis-ease....

Response 6: We appreciate the comment.  We have added abbreviation and corrected the grammar in line 60.

Commented [M7]: How do we know that??? Reference(s)???

Response 7:

We appreciate the inquiry. Pancreatic ductal adenocarcinoma (PDAC) exhibits significant genetic diversity and heterogeneity, characterized by numerous mutations in key oncogenes and tumor suppressor genes. Whole-genome sequencing has identified critical genetic defects and complex mutational landscapes in PDAC, which are essential for understanding the disease's biology and developing targeted therapies, as seen in reference [9]. Detailed genetic profiling facilitates the development and application of targeted therapies tailored to the unique genetic makeup of an individual’s cancer.

Nearly all PDACs harbor activating mutations in the KRAS gene, which drives tumor growth. While direct targeting of KRAS has been challenging, downstream signaling pathways (such as MEK and ERK) are being explored for therapeutic intervention. Germline and somatic mutations in BRCA1 and BRCA2 are found in a subset of PDAC patients. These mutations are linked to defects in DNA repair mechanisms. Patients with these mutations have shown responsiveness to DNA-damaging agents like platinum-based chemotherapy and PARP inhibitors, which exploit the tumor's inability to repair DNA damage effectively. Some PDACs exhibit amplification of the HER2 gene, and targeting HER2 with monoclonal antibodies (such as trastuzumab) or tyrosine kinase inhibitors can be an effective strategy for these patients.

We have added a reference addressing all of this [11] in line 69 that could potentially help understand our point better.

Commented [M8]: How did the Authors determine the study size? Protocol? Reference(s)? Inclusion / Ex-clusion Criteria?

Response 8:

To determine if our sample size of 142 patients is adequate for meaningful statistical analyses, we considered several key factors. For Kaplan-Meier survival curves and Cox proportional hazards regression, we aimed for a power (1 - β) of at least 0.80 and a significance level (α) of 0.05. These criteria ensure that our study can detect significant differences in survival rates with high confidence.

We used an established formula to estimate the required sample size for survival analysis, which considers the critical values for our chosen significance level and power, the proportion of events observed in the study, and the minimum detectable difference in survival rates between groups. Specifically, we used the critical values for a significance level of 0.05 (1.96) and for a power of 0.80 (0.84). We assumed an event rate of 50%, a common assumption in survival studies, and aimed to detect a difference in survival rates of at least 20% between groups.

Based on these parameters, the sample size calculation indicated that 49 patients per group would be required to detect a significant difference in survival rates. Given that our study includes multiple groups and stratifications based on oncogenic pathways and treatment regimens, a larger overall sample size is necessary to maintain statistical power across these comparisons. Thus, the final sample size of 142 patients was deemed adequate. This size allows for meaningful subgroup analyses and maintains statistical power for detecting significant differences in survival outcomes and potential predictive biomarkers.

After determining the necessary sample size, we collected data from patients meeting the inclusion criteria at the medical oncology clinic of Beth Israel Deaconess Medical Center (BIDMC) between 2013 and 2022. Our goal was to create a comprehensive dataset of patients with unresectable pancreatic ductal adenocarcinoma (PDAC) who underwent next-generation sequencing (NGS) and cytotoxic chemotherapy. This approach ensures that our study is well-powered to explore the relationship between genetic mutations, treatment responses, and survival outcomes in this patient population.

Inclusion Criteria:

  • Patients diagnosed with unresectable PDAC (Stage III or IV).
  • Patients who underwent NGS for molecular profiling.
  • Patients who received first-line systemic cytotoxic chemotherapy (FOLFIRINOX or gemcitabine-nab paclitaxel).

Exclusion Criteria:

  • Patients with resectable PDAC.
  • Patients without available NGS data.
  • Patients who did not receive cytotoxic chemotherapy as their first-line treatment.

We have corrected the text to reflect these changes to emphasize this point from line 84 to line 92.

Commented [M9]: Based on which criteria? Refer-ence(s)???

Response 9: We appreciate the comment. Unresectable pancreatic cancer was defined according to the criteria outlined in the Sixth Edition of the American Joint Committee on Cancer (AJCC) for stages III and IV. Added reference.

Commented [M10]: Did the Authors use a standard-ized questionnaire for those variables examined? If so, state Reference(s).....

We performed a retrospective chart review; therefore, a standardized questionnaire was not needed. This also ensured we did not need any direct contact with patients or clinicians. This methodology allowed us to gather comprehensive data from existing medical records.

Commented [M11]: Reference(s)?????

Response 11: We have added the reference

Commented [M12]: Reference(s)?????

Response 12:  We have added the reference

Commented [M13]: "studies" or "a study"???????

Response 13: Thank you for pointing this out. We meant studies and added citation

Commented [M14]: Methodological Biases exist...

Thank you for your review. We addressed this issue in Comment 7 by providing a detailed report on potential risks for biases and how we mitigated them.

Commented [M15]: References????

Response 15: Thank you for your comment. We have addressed this concern as follows: We agreed with your comments on line 472 regarding the need for references. Our pathway classification was based on the Reactome Pathway Database, and we have added this reference at the beginning of the table. Additionally, we have provided specific references for each pathway classification. These citations have been included in the revised version of the manuscript.

Reviewer 2 Report

Comments and Suggestions for Authors

The submitted manuscript presents a short communication on the somatic mutation profile and its possible applications as a predictor of treatment response and survival rate in unresectable PDAC. The study is a simple retrospective analysis, on a small number of patients, from a single institution. The level of novelty is therefore quite low. However, the statistical-analytical methods have been chosen and used properly. Besides, there are also some problems that need to be addressed, I’ve listed them below.

Major issues:

Line 22, it’s not clear for me how can a retrospective study improve survival. It can provide some suggestions that need to be verified clinically. I advise to add “possibly” as a first word in Line 22

Reference [1] should be replaced by more recent one.

Lines 44-45, why this option is so limited?

Lines 49, here the most common therapeutic options should be discussed in more details.

Tables 2 and 3, is there a reason why some rows are bolded?

Lines 442-443, the Authors haven’t uploaded any supplementary materials…

At the end of the manuscript the Authors should create the list of abbreviations. I know this is not mandatory, but the Authors have used a lot of them in the current study. Therefore, this would be beneficial.

Minor comments:

Lines 9-18, shouldn’t it be a single affiliation since they are all the same?

Table 1, I guess “UNK” is unknown?

Lines 360, 366, 369, the Authors keep writing “in our study”, which should not be used do often; rather replace with “in the current study” “in the current work/analysis”

Author Response

For research article

Response to Reviewer X Comments

1. Summary

Thank you very much for taking the time to review this manuscript. We appreciate your valuable feedback and suggestions. Please find our detailed responses below and the corresponding revisions/corrections highlighted in the re-submitted files. Your insights have been instrumental in improving the quality of our work.

2. Point-by-point response to Comments and Suggestions for Authors

Comment 1 : The submitted manuscript presents a short communication on the somatic mutation profile and its possible applications as a predictor of treatment response and survival rate in unresectable PDAC. The study is a simple retrospective analysis, on a small number of patients, from a single institution. The level of novelty is therefore quite low. However, the statistical-analytical methods have been chosen and used properly. Besides, there are also some problems that need to be addressed, I’ve listed them below.

Response 1:  Thank you for your critical review of our manuscript. We appreciate your acknowledgment of our statistical-analytical methods. While we understand your concern regarding the study's scope and the number of patients, we believe that our work offers valuable insights into the molecular profiling of unresectable PDAC. This study presents a new dataset that links mutational pathways as predictors of treatment response in PDAC, an area where such data is sparse and further research is needed.

Although the population was small, the balanced characteristics of the cohort provide better insight into how specific genetic mutations can predict treatment outcomes. Our analysis highlighted the potential for molecular profiling to guide personalized treatment strategies and contribute to the growing body of evidence in this field.

We have acknowledged the limitations of our study in the manuscript, including the small sample size and single-institution setting, in our discussion section and emphasized the need for further multi-institutional studies with larger sample sizes to validate our findings.

Major issues:

Comments 2: Line 22, it’s not clear for me how can a retrospective study improve survival. It can provide some suggestions that need to be verified clinically. I advise adding “possibly” as the first word in Line 22.

Response 2: Thank you for your input. We agree with your comment and have revised the line to: "This study aims to show how molecular profiling could possibly guide personalized treatment strategies, which may help improve survival outcomes in patients with PDAC," as documented in line 23.

Comments 3: Reference [1] should be replaced by a more recent one.

Response 3: We agree with this comment and have updated our reference. Pancreatic cancer has raised to the third-leading cause of cancer death in men and women combined. We reflected this change in our manuscript in line 44.

Comments 4: Lines 44-45, why is this option so limited?

Response 4: We added a line to clarify why the surgical options are limited. In general, most PDACs are found at a later stage when surgery is not an option anymore.

Comments 5: Lines 49, here the most common therapeutic options should be discussed in more detail.

Response 5: We agree with the comment. We added an additional line explaining the most common therapies.

Comments 6: Tables 2 and 3, is there a reason why some rows are bolded?

Response 6: Those were the ones that were statistically significant, which is why we wanted to highlight them. We un-bolded them to maintain consistency.

Comments 7: Lines 442-443, the Authors haven’t uploaded any supplementary materials…

Response 7: We apologize; this line was not deleted from the original format provided.

Comments 8: At the end of the manuscript, the Authors should create a list of abbreviations. I know this is not mandatory, but the Authors have used a lot of them in the current study. Therefore, this would be beneficial.

Response 8: We appreciate your input. We decided to include a new appendix (Appendix B) with the abbreviations used.

Minor comments:

Comments 9: Lines 9-18, shouldn’t it be a single affiliation since they are all the same?

Response 9: In the original format, we were asked to provide individual emails for authors, each of which needed to include an affiliation. This is why the affiliations were repeated. However, we have now changed it to list only the institutions and included the contact information for the corresponding authors.

Comments 10: Table 1, I guess “UNK” is unknown?

Response 10: Thanks for the feedback. We wrote unknown for clarity.

Comments 11: Lines 360, 366, 369, the Authors keep writing “in our study”, which should not be used so often; rather replace with “in the current study” “in the current work/analysis”

Response 11: Thank you for your suggestion. We have revised the text to replace 'in our study' with 'in the current study,' 'in the current work,' or 'in the current analysis' as appropriate.

3. Additional clarifications

We have provided a document with track changes as well as the corrected manuscript with the changes highlighted in red to better illustrate the revisions made. This should help you easily identify and review the modifications we have implemented in response to your comments

Round 2

Reviewer 1 Report

Comments and Suggestions for Authors

All is in order!!!

Reviewer 2 Report

Comments and Suggestions for Authors

The Authors have provided sufficent answers to my questions and corrected their work accordingly. Current version can be accepted for publication.